# Radiocarbon-Refined Archaeological Chronology and the History of Human Activity in the Southern Tarim Basin

Xiaofang Ma [1], Xingjun Hu [2,3] and Menghan Qiu [4,*]

1 Research Institute of Specialized History, School of History and Culture, Lanzhou University, Lanzhou 730000, China; 120220906460@lzu.edu.cn
2 Research Center for Governance of China's Northwest Frontier in the Historical Periods, School of History, Xinjiang University, Urumqi 830046, China; 13579886217@163.com
3 Xinjiang Institute of Cultural Relics and Archaeology, Urumqi 830011, China
4 MOE Key Laboratory of Western China's Environmental Systems, Collage of Earth and Environmental Sciences, Lanzhou University, Lanzhou 730000, China
* Correspondence: choumh14@lzu.edu.cn

**Abstract:** Famous for Taklimakan, the world's second largest sandy desert, the Tarim Basin in Xinjiang has long attracted researchers from various fields to investigate its paleoenvironment and antiquity. The southern part of this basin is an ideal region in which to investigate the interactions between humans and the environment due to its fragile habitat and prosperous ancient civilizations. However, the lack of direct radiocarbon dating data has caused the chronologies of some of the archaeological sites to be debatable, which hinders our ability to reconstruct historical patterns of human activity and further understand, in a coherent manner, their interaction with the environment. This study reports 25 new radiocarbon dates acquired from ten undated archaeological sites in the southern Taklimakan Desert in order to refine their chronologies. Based on this, a radiocarbon dataset was established to reveal the trajectory of human activity with the support of Bayesian chronological modeling. The results indicate a two-millennium continuous flourishing of the local society since the beginning of the first millennium BCE, as well as a peak of human activity during the Tang Dynasty (618–907 CE). The distinct trajectory of human activity in the southern Tarim Basin revealed by this study provides a solid foundation for further assessments of human–environment interaction in the Tarim Basin and along the Silk Road.

**Keywords:** the Silk Road; arid region; Taklimakan Desert; Bayesian analysis; social evolution

## 1. Introduction

Arid Central Asia is witnessing an increasingly tense relationship between humans and the environment against the background of global change due to its fragile ecological environment and sensitivity to climate change [1]. Studies on historical interactions between humans and the environment are key to facing the challenges of today and predicting the future [2]. The Tarim Basin, located in Xinjiang, China, has become a hotspot for revealing historical human–environment interactions due to its extremely arid climate, independent geographic setting, and abundant antiquities [3–6].

Recent decades have witnessed abundant research on climatic and environmental changes, historical human activities, and their interactions in the Tarim Basin. Benefiting from a proper timescale and high-quality resolution, lacustrine sediments from lakes such as Bosten [6], Swan [7], Kalakuli [8], and Lop Nur [9] provide credible reconstructions of the environmental background of human activity. Progress in speleothem and tree-ring studies over recent years has revealed new insights into the climate in order to interpret the region's historical human–environment interactions [10,11]. An increasing number of local paleoenvironmental reconstructions are becoming available to enable the better

understanding of historical human activity patterns [12–14]. Archaeological and historical research has established a framework for the history of human activity in the Tarim Basin. In addition to the discovery of a number of representative prehistoric cultures such as Xiaohe [15] and Chawuhu [16], multidisciplinary research has preliminarily revealed information about subsistence, migration, and exchange in Bronze Age and Early Iron Age societies [17–24]. Issues surrounding the ancient kingdoms along the Silk Road remain a focus of historical studies [1,25,26]. Some of the most recent research on archaeological sites in Aketala-Wupaer [27,28], North Loulan [29], Kuiyukexiehai'er [30], Keyakekuduke [31,32], Haermodun [33], and Miran [34] has provided new perspectives for the study of agricultural development, settlement evolution, and East–West communications along the Silk Road. Despite the intense focus on human–environment interactions in the northern part of the Tarim Basin, less attention has been paid to the south.

In contrast to the vast oasis strung by the Tarim River in the north, the oases of the southern Tarim Basin are relatively small and mutually independent, which may have restricted early trans-regional communication and the scale of human activities. Due to the absence of lacustrine sediments, the reconstruction of a historical environmental background of human activity has mainly focused on eolian deposits and paleo-oases, which has shed some light on the interaction between oasis evolution and human activities [35,36]. A conceptual model describing the relations between the climate, hydrology, and society of an arid mountain-basin system has also improved our understanding of the complex human–environment interactions in arid regions [37]. In the domain of archaeology, the second and third national surveys of cultural relics revealed dozens of archaeological sites in the southern Tarim Basin [38,39], the findings of which have become a primary reference for relevant research [40–43]. Despite the fact that researchers have dated the activity of Paleolithic human groups in the Kunlun Mountains during the Middle Holocene [44], it is still possible that it was not until the second half of the second millennium BCE that sedentary populations emerged in the oases [45]. Researchers have conducted a series of archaeological investigations and excavations in the vast area stretching from the mountain valleys to the hinterland of the Taklimakan Desert in the past few decades, and have yielded dozens of radiocarbon dates [17,36,44–49]. These efforts contribute to much of the understanding of the region's antiquity. However, only a few of these radiocarbon dates were measured using accelerator mass spectrometry (AMS). Furthermore, the application of radiocarbon dating was mainly concentrated at several specific prehistoric sites. Because historical documentation of the southern Tarim Basin is relatively scarce and fragmented, it is not easy to identify the age of some of its historical sites. Therefore, it is necessary to carry out more precise AMS radiocarbon dating when refining the absolute chronology of archaeological sites of different historical periods.

This study aims to provide a more solid chronological reference for a batch of undated archaeological sites in the southern Tarim Basin using AMS radiocarbon dating. In addition, a regional radiocarbon dataset is to be established under the restriction of Bayesian chronological modeling to refine the chronology of these well-dated archaeological sites and to reveal the trajectory of human activity. This more refined local history may pave the way towards a better understanding of human–environment interactions along the Silk Road.

## 2. Study Area

Situated in the arid region of middle Eurasia, the Tarim Basin is a unique geographic unit surrounded by the Tianshan Mountains in the north, the Pamir Plateau in the west, and the Kunlun Mountains in the south (Figure 1a). With some of the peaks reaching altitudes of 7–8 km, these towering mountains notably block the westerlies and prevent moisture from entering the basin, thus bringing an extremely arid climate to the Tarim Basin, and forming the world's second largest sandy desert, Taklimakan. However, these mountains are collectors of atmospheric moisture, aiding in the formation of vast forest and steppe areas, as well as providing massive runoff for the surrounding arid lowlands. Water

runoff in the Tarim Basin forms several major rivers and finally converges into Lop Nur in the east. The severe disequilibrium of water resources restricts lowland human activity to a certain extent within the ecological hotspots, i.e., the oases. In the southern Tarim Basin, rivers originating from the Kunlun Mountains in the south flow freely north into the Taklimakan Desert, creating oases along the margins of their fluvial fans and along their channels, and have potentially shaped the patterns of human activity from the past to the present (Figure 1b). The Silk Road, which was formally opened during the Han Dynasty (202 BCE–220 CE), spread its south route in Xinjiang through these oases and settlements, and witnessed the flourishing human activity in this region.

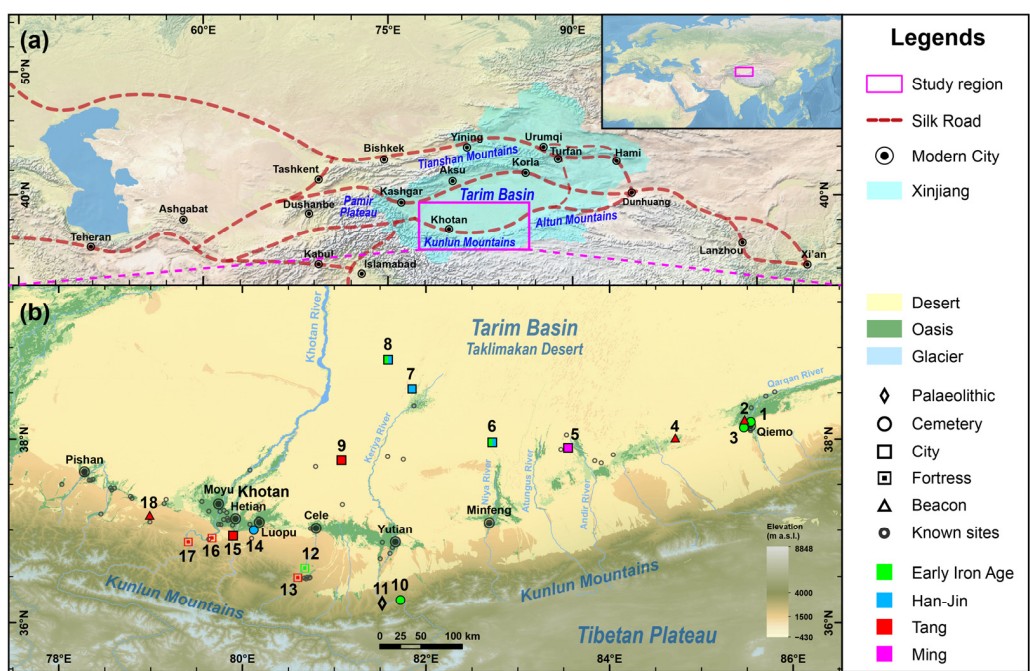

**Figure 1.** Maps of the study region. (**a**) A general map showing the geographic settings of the southern Tarim Basin and Xinjiang in arid Central Asia (base map sourced from Natural Earth Data: https://www.naturalearthdata.com/ (accessed on 10 January 2024)). (**b**) Geographic background and archaeological sites of the southern Tarim Basin (Digital Elevation Model sourced from https://www.gscloud.cn/ (accessed on 10 January 2024)). Archaeological sites with radiocarbon dates are labeled as (1) Jiawaairike cemetery; (2) Jiandatierimu beacon; (3) Zhagunluke cemetery; (4) Bugunluke beacon; (5) Akekaoqikaranke ancient city (AKKQKRK); (6) Niya ancient city; (7) Kaladun ancient city; (8) Yuansha ancient city; (9) Dandanwulike ancient city (DDWLK); (10) Liushui Cemetery; (11) Yangchang Paleolithic site; (12) Axi fortress; (13) Asa fortress; (14) Shanpula cemetery; (15) Mailikeawati ancient city; (16) Aqike fortress; (17) Puji fortress; and (18) Duwaxi beacon.

## 3. Materials and Methods

### 3.1. Field Investigation and Sampling

Aiming to collect organic samples with reliable stratigraphic information from a selection of archaeological sites in order to yield robust radiocarbon dates, we carried out a field investigation in 2018–2019 (Figure 2) based on the findings reported by the second and third national surveys of cultural relics [38,39]. Organic samples, such as architectural relics and artificial objects buried in cultural layers, which can represent the date of formation and/or utilization, were collected. Short-lived materials were preferentially selected for radiocarbon dating to reduce the potential influence of the old carbon effects [50,51].

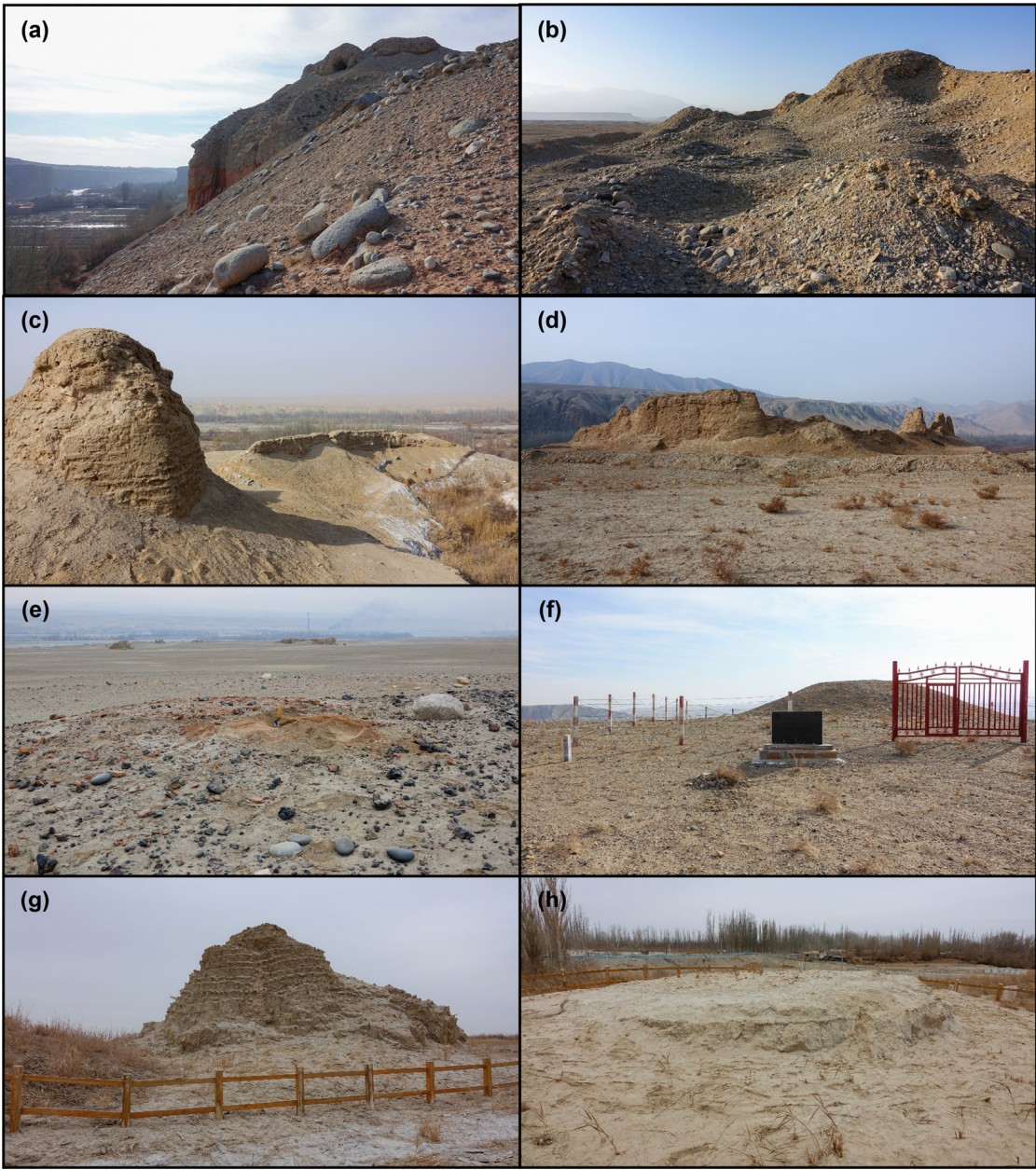

**Figure 2.** Photographs of some of the investigated archaeological sites: (**a**) Asa fortress; (**b**) Axi fortress; (**c**) Aqike fortress; (**d**) Puji fortress; (**e**) a kiln at the Mailikeawati ancient city; (**f**) Duwaxi beacon; (**g**) Bugunluke beacon; (**h**) Jiandatierimu beacon.

### 3.2. Radiocarbon Dating

AMS radiocarbon dating was carried out at the Key Laboratory of Western China's Environmental Systems (Ministry of Education) in Lanzhou University using an MICADAS system. Details about the laboratory and the relevant protocols were reported in a recent paper [52]. The OxCal 4.4.4 program and the IntCal20 calibration curve were used in this study to calibrate the original radiocarbon dates [53–55]. We also established a radiocarbon dataset of the southern Tarim Basin to generate a curve for the summed probability distribution (SPD) using the same program via the "Sum" function, as this curve is more frequently used to illustrate the intensity of human activity in paleodemographical studies [56–59].

### 3.3. Bayesian Chronological Modeling

Bayesian chronological modeling has been widely applied to restrict the age boundaries of clusters of radiocarbon dates in the research of archaeological chronology [53,60–63]. The methods of this study were based on previous protocols, and the radiocarbon dates that were distinct from the rest of the data group were judged as outliers and were eliminated before modeling [60–62]. Bayesian chronological modeling was applied to restrict the chronologies of some of the well-dated archaeological sites in the southern Tarim Basin, taking the median-to-median age from the two boundaries as their time span.

## 4. Results and Discussion

### 4.1. New Radiocarbon Dates

This study yielded 25 AMS radiocarbon dates from ten archaeological sites in the southern Tarim Basin (Table 1). Detailed information of these new data and other published data were collected in the radiocarbon database, as is shown in Table S1. The calibrated radiocarbon dates span from 760 cal. BCE to 1660 cal. CE, covering the extensive period from the Early Iron Age to the Ming–Qing Dynasties (1368–1911 CE). However, most of the radiocarbon dates concentrate in or around the Sui–Tang period (581–907 CE), indicating an increase in the number of archaeological sites and the rise of human activities during this period.

**Table 1.** New radiocarbon dates reported in this study. BP stands for years before present (taken as 1950 CE).

| Site Name | Site Type | Material Dated | Lab Code | $^{14}$C Age (BP) | Uncertainty (Year) | Calibrated Age 2σ |
|---|---|---|---|---|---|---|
| AKKQKRK | City | Branch | LZU19091 | 285 | 20 | 1520–1660 CE |
| | | Branch | LZU19092 | 335 | 20 | 1480–1640 CE |
| | | Charcoal | LZU19093 | 535 | 20 | 1320–1440 CE |
| | | Branch | LZU19094 | 325 | 20 | 1490–1640 CE |
| | | Charcoal | LZU19095 | 410 | 25 | 1430–1620 CE |
| | | Branch | LZU19096 | 310 | 25 | 1490–1650 CE |
| Aqike | Fortress | Straw | LZU19112 | 1420 | 25 | 590–660 CE |
| | | Branch | LZU19113 | 1410 | 20 | 600–660 CE |
| Asa | Fortress | Wood | LZU19103 | 1285 | 25 | 660–780 CE |
| | | Wood | LZU19104 | 1305 | 25 | 650–780 CE |
| Axi | Fortress | Wood | LZU19105 | 2465 | 25 | 760–420 BCE |
| | | Branch | LZU19106 | 2390 | 25 | 550–390 BCE |
| Bugunluke | Beacon | Wood | LZU19089 | 1245 | 25 | 670–880 CE |
| | | Wood | LZU19090 | 1240 | 20 | 680–880 CE |
| DDWLK | City | Reed | LZU19097 | 1230 | 25 | 680–890 CE |
| | | Reed | LZU19098 | 1275 | 25 | 660–820 CE |
| | | Fabric | LZU19099 | 1055 | 30 | 890–1040 CE |
| | | Reed | LZU19100 | 1225 | 25 | 690–890 CE |
| | | Straw | LZU19101 | 1285 | 30 | 660–820 CE |
| | | Wood | LZU19102 | 1185 | 20 | 770–900 CE |
| Duwaxi | Beacon | Branch | LZU19107 | 1080 | 25 | 890–1030 CE |
| Jiandatierimu | Beacon | Straw | LZU19087 | 1085 | 30 | 890–1030 CE |
| Mailikeawati | City | Charcoal | LZU19109 | 1360 | 40 | 600–780 CE |
| Puji | Fortress | Wood | LZU19110 | 1440 | 25 | 580–660 CE |
| | | Branch | LZU19111 | 1460 | 25 | 570–650 CE |

The archaeological and geographic information of all archaeological sites examined in this study is listed in Table 2. The ages of the archaeological sites dated in this study were previously determined according to the second national survey of cultural relics [38]. The results of the radiocarbon dating are generally consistent with previous judgments, but still suggest discrepancies in the chronology of some of the archaeological sites, which are labeled with asterisks in Table 2.

**Table 2.** Archaeological sites with radiocarbon dates. Sites with chronology different from previous estimation are labeled with an asterisk. Abbreviation: PA, Paleolithic Age; BA, Bronze Age; EIA, Early Iron Age; WJSN, Wei, Jin, Southern and Northern Dynasties; FDTK, Five Dynasties and Ten Kingdoms.

| Site Name | Site Type | Cultural Period | Administration | Latitude (° N) | Longitude (° E) | Altitude (m) | Data Source |
|---|---|---|---|---|---|---|---|
| AKKQKRK * | City | Ming | Minfeng | 37.914 | 83.548 | 1218 | This study. |
| Aqike | Fortress | Tang | Hetian | 36.930 | 79.668 | 1520 | This study. |
| Asa * | Fortress | Tang | Cele | 36.508 | 80.603 | 2144 | This study. |
| Axi * | Fortress | EIA | Cele | 36.607 | 80.680 | 1953 | This study. |
| Bugunluke * | Beacon | Tang | Qiemo | 38.018 | 84.715 | 1211 | This study. |
| DDWLK | City | Tang | Cele | 37.776 | 81.079 | 1215 | This study. |
| Duwaxi * | Beacon | FDTK | Moyu | 37.172 | 78.989 | 1688 | This study. |
| Jiandatierimu * | Beacon | FDTK | Qiemo | 38.199 | 85.509 | 1183 | This study. |
| Mailikeawati | City | Tang | Hetian | 36.952 | 79.899 | 1467 | This study. |
| Puji | Fortress | Tang | Hetian | 36.886 | 79.410 | 1804 | This study. |
| Jiawaairike | Cemetery | EIA–Han | Qiemo | 38.169 | 85.525 | 1244 | [46] |
| Kaladun | City | Han | Yutian | 38.558 | 81.848 | 1200 | [47] |
| Liushui | Cemetery | EIA | Yutian | 36.245 | 81.723 | 2850 | [17] |
| Niya | City | EIA–WJSN | Minfeng | 37.976 | 82.721 | 1250 | [48] |
| Shanpula | Cemetery | EIA–Han | Luopu | 36.999 | 80.124 | 1407 | [49] |
| Yangchang | Profile | PA | Minfeng | 36.219 | 81.521 | 2440 | [44] |
| Yuansha | City | EIA–WJSN | Yutian | 38.871 | 81.582 | 1200 | [36,47] |
| Zhagunluke | Cemetery | BA–WJSN | Qiemo | 38.121 | 85.475 | 1270 | [45] |

*4.2. Refined Chronology of the Archaeological Sites*

Situated in the tailwater zone of the Andir River and currently surrounded by the sand dunes of the Taklimakan Desert (Figures 1b and 3a), the ancient city AKKQKRK was previously judged to be coeval with the Song Dynasty in Central China (960–1368 CE) [38]. However, the result of this study shows that all six radiocarbon dates acquired from different locations of the site largely fall into the period of Ming Dynasty (1368–1644 CE), and the Bayesian-restricted time span of the site is 1360–1640 cal. CE (Figure 3a). There seems to be a discrepancy between the finding in this study and that obtained from the previous estimation. A possible reason for the previous misjudgment could be attributed to the lack of typical material relics and historical documentation.

The DDWLK (or Dandan Oilik) site is another ancient city that was well dated in this study. Underpinned on several Yardang tablelands, the site is currently covered by mobile sand dunes (Figure 3b). First investigated by explorer Sven Hedin in 1895–1896 CE, and re-investigated in detail by the Sino-Japan Joint Investigation Team in the first decade of the 21st century CE, the site is thought to be that of the Jiexie Township of the Yutian Kingdom (232 BCE–1006 CE) during the Tang Dynasty [64,65]. We acquired six organic samples from five different locations at the site, including the city wall, a residential dwelling, and the wall of a herd enclosure. The Bayesian-restricted age of the site falls into 660–1010 cal. CE (Figure 3d). This finding is in general agreement with previous understandings. The result also suggests that the site seems to have synchronously declined in 1006 CE when the Yutian Kingdom was conquered by the Karakhanid Dynasty [66].

Asa and Axi are two wall-fortified sites situated on the cliffs above the upper branches of the Cele River (Figure 2a,b). The Asa fortress was previously known as the last stand for the Yutian people to resist invasion after the vanish of their kingdom according to the tales spreading among local villagers [67]. Thus, both sites were previously thought to be fortresses of the Song Dynasty in Central China [38]. However, our results shed some new light on the two sites, although there is less data than for AKKQKRK and DDWLK. Each of the fortresses yielded two similar radiocarbon dates from their walls. The age of the Asa fortress falls into the earlier half of the Tang Dynasty (Table 1), centuries earlier than that judged by the previous estimation. Considering that these two samples were collected

from the logs inserted into the wall, the potential influence of the old wood could not be excluded [50,51]. Further excavation and more fine-grained dating may help to restrict the chronology. More intriguingly, the Axi fortress was traced back to 760–390 cal. BCE (2σ range), in accordance with our findings, which falls into the Early Iron Age (Table 1). Mainly built up with mud and pebble stones, the wall construction of the Axi fortress is indeed different from that of the commonly seen earth-rammed walls throughout the historical period of Xinjiang (Figure 2b). Sites found in the Eastern Tianshan Mountains that share a similar geographic setting and archaeological features are usually identified as settlements of the Early Iron Age pastoralists [68]. Moreover, the Liushui cemetery of the early-first-millennium BCE pastoralists was excavated in a river valley adjacent to the Axi site, and has been well reported [17,69,70]. Results from a physical anthropology study also indicated a close relation between the Liushui cemetery and the Tianshanbeilu cemetery in the Eastern Tianshan Mountains [71]. This evidence suggests that the Axi fortress could possibly be a settlement of the Early Iron Age pastoralists in the Kunlun Mountains.

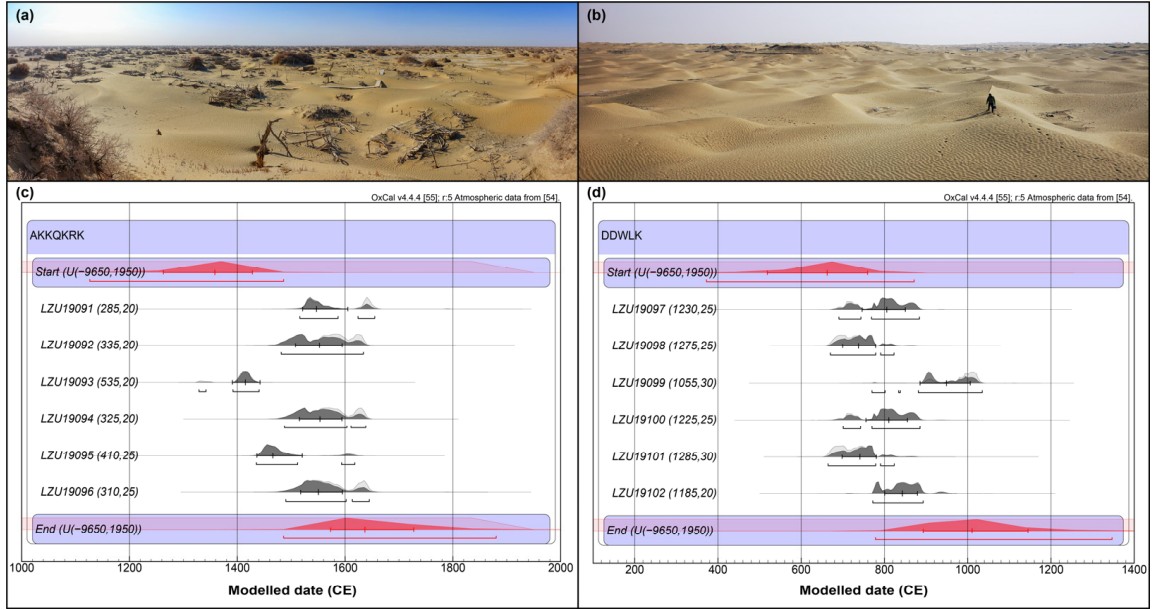

**Figure 3.** Landscape and chronology of AKKQKRK and DDWLK sites: (**a**) panoramic photography showing the landscape at AKKQKRK; (**b**) panoramic photography showing the landscape at DDWLK; (**c**) radiocarbon dates of AKKQKRK restricted by Bayesian statistic model; (**d**) radiocarbon dates of DDWLK restricted by Bayesian statistic model. All radiocarbon dates (labelled in grey) were calibrated by IntCal20 [54] and the boundaries of the Bayesian statistic modeling (labelled in red) were generated using the OxCal 4.4.4 program [55].

The three beacons, Bugunluke, Jiandatierimu, and Duwaxi, were also provided with refined chronologies in this study (Tables 1 and 2). As a recently found site, the Bugunluke beacon was dated to the Tang Dynasty, indicating it as a node of the Tang military towns' defense system. Previously, the Jiandatierimu and Duwaxi beacons were dated to the Han Dynasty and Tang Dynasty, respectively. This study traced them both back to the FDTK period (907–979 CE) (Table 1), suggesting that these two beacons might have been built or reinforced by the local regimes rather than the Tang government.

In addition to the new radiocarbon dates, Bayesian modeling was also applied to reassess the chronology of four previously dated archaeological sites: Liushui cemetery, Niya ancient city, Shanpula cemetery, and Yuansha ancient city (Figure 4; Table S1). The Bayesian-restricted chronology of the previously mentioned Liushui cemetery falls into 990–700 cal. BCE (Figure 4a) [17]. Known as the Jingjue Kingdom on the south route of Xinjiang's Silk Road, the ancient city and the surrounding dwellings of Niya were

previously investigated and well reported [72–75]. The collapse and abandonment of the kingdom were broadly discussed and attributed to three distinct reasons: (1) geopolitics and warfare; (2) environmental change and desertification; or (3) a great earthquake [76,77]. Although the collapse of the kingdom was thought to have occurred much earlier [77], the radiocarbon dates suggest that human activity in the region lasted until the seventh century CE, with a southward shift with time [48]. The Bayesian-restricted chronology of the site falls into 510 cal. BCE–660 cal. CE, indicating a long period of human activity in the tailwater zone of the Niya River (Figures 1b and 4b). The time span of the utilization of Shanpula, a cemetery of the Yutian Kingdom, is relatively shorter compared to the long-lived Niya site [49]. With one notably later date eliminated, the rest of the radiocarbon dates were examined with Bayesian modeling and fall into 220 cal. BCE–10 cal. CE (Figure 4c). Located beside the ancient channel of the Keriya River, the ancient city Yuansha is another well-dated site, known as the capital of the Yumi Kingdom (Figure 1b). Similar to the Niya site, historical documentation also revealed a southward relocation of Yumi's capital city from Yuansha to Kaladun, which was also attributed to environmental change and a lack of water resources [78]. Radiocarbon dates acquired directly from artificial remains were collected for Bayesian modeling, and the data analyses indicate a period of human activity spanning from 520 cal. BCE to 440 cal. CE for the region of Yuansha (Figure 4d).

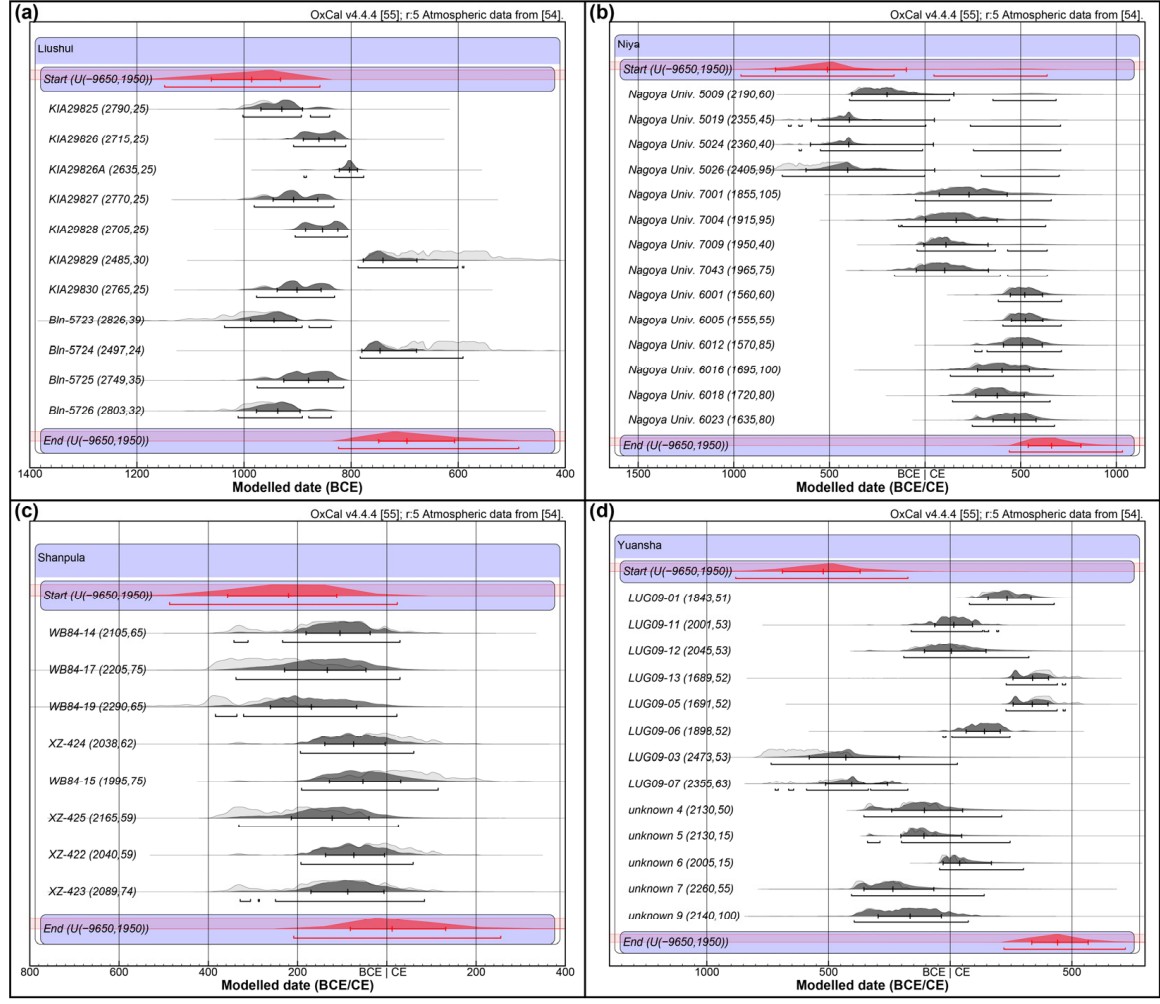

**Figure 4.** Chronology of previously dated archaeological sites restricted with a Bayesian statistic model: (**a**) Liushui cemetery [17]; (**b**) Niya ancient city [48]; (**c**) Shanpula cemetery [49]; (**d**) Yuansha ancient city [36,47]. All radiocarbon dates (labelled in grey) were calibrated by IntCal20 [54] and the boundaries of the Bayesian statistic modeling (labelled in red) were generated using the OxCal 4.4.4 program [55].

*4.3. A Clearer Local History of Human Activity and Its Implication for Human–Environment Interactions*

Constructed with 80 radiocarbon dates, the SPD curve of the southern Tarim Basin reveals a continuous flourishing of the local society from ca. 1000 BCE to 1000 CE (Figure 5; Table S1). Radiocarbon dates from the Yangchang Paleolithic profile [44] were excluded for the purpose of focusing more on the agro-pastoral society. Archaeological evidence and the radiocarbon curve indicate that nomadic pastoralists occupied the grasslands in front of the Kunlun Mountains since the beginning of the first millennium BCE, incorporating this region into the pastoral areas of Early Iron Age middle Eurasia [17]. Meanwhile, sedentary society also sprouted on the oases of the southern Tarim Basin [45,46]. From there, city states and/or kingdoms, such as Yutian, Qiemo, Jingjue, and Yumi, began to develop with the rising economy and increasing population, gradually becoming interconnected with other regions of the Tarim Basin, which laid the foundation for the opening of the Silk Road [79].

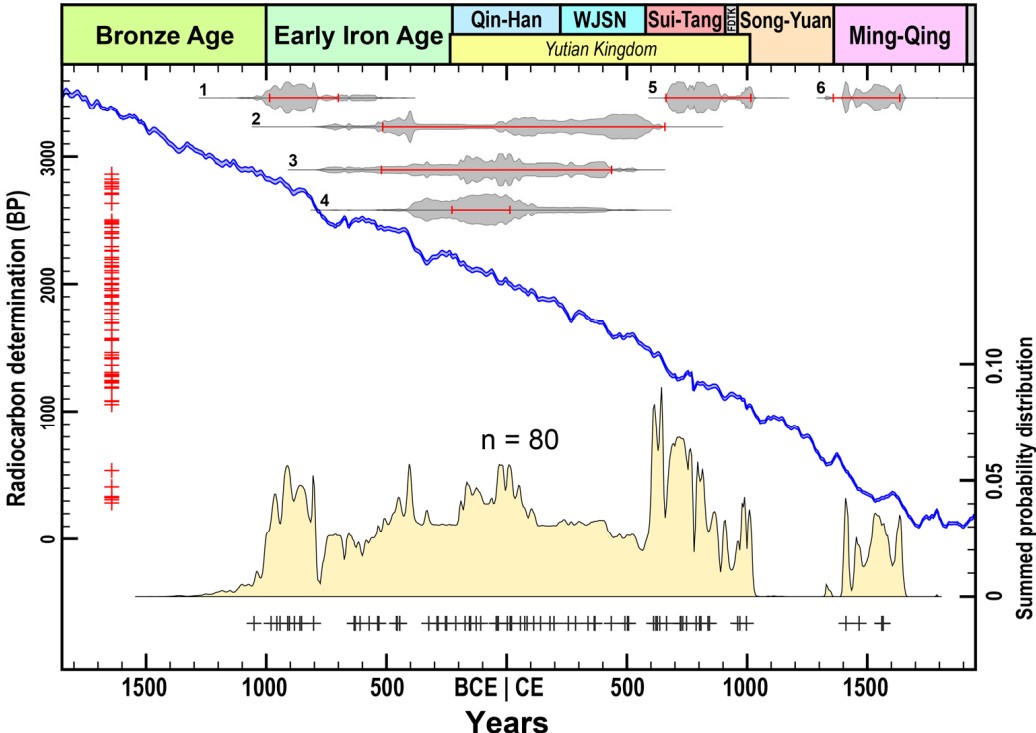

**Figure 5.** Chronology of the archaeological sites in the southern Taklimakan Desert indicated by SPD (curve with yellow shading) of 80 radiocarbon dates (Table S1) and the refined chronology of key archaeological sites labeled as (1) Liushui cemetery; (2) Niya ancient city; (3) Shanpula cemetery; (4) Yuansha ancient city; (5) DDWLK; (6) AKKQKRK. The shadows in grey represent the SPD of each site; crosses in red and black represent the original and calibrated radiocarbon dates, respectively, while the red bars represent the chronology restricted by Bayesian chronological modeling. Radiocarbon curves were generated using the OxCal 4.4.4 program [55].

The time spanning from the Qin–Han (221 BCE–220 CE) to Sui–Tang (581–907 CE) period witnessed a southward shrinkage in the area of residence along various rivers, although human activity in the southern Tarim Basin continued to thrive (Figure 1b) [36,78]. Due to land degradation, settlements in the hinterland of the Taklimakan Desert were abandoned during the Tang Dynasty [3]. A similar pattern of human activity was also detected in the eastern Tarim Basin, which is especially famous for the decline of the Loulan Kingdom [80]. Inversely, the overall human activity reached a peak, not only in southern Tarim Basin (Figure 5), but also in the whole region of Xinjiang's Silk Road [43]. Given that land degradation and the shrinkage of the oases and tailwater have been generally

validated [3,4,36,81], different hypotheses were developed to explain the collapse of ancient civilizations in the hinterland of the Taklimakan Desert: (1) environmental deterioration due to climate change [43,80,82]; (2) environmental deterioration caused by excessive tillage and irrigation [5,78]; and (3) the impact of geopolitics and frequent warfare due to the increasing living pressure [77]. Despite the main reason for the decline in civilization not being well proven, almost all hypotheses illustrate the importance of water resources in the Tarim Basin for maintaining a habitable local environment and stable ancient society. Hence, reasonable management and utilization of water resources is always the key to achieving sustainable development, not only in the past, but also in the future. Although the refined chronology of the southern Tarim Basin sketches its trajectory of human activity, further detailed research from both environmental and archaeological perspectives is needed to achieve a more comprehensive understanding.

Human activity in the southern Tarim Basin began to decline during the FDTK period (Figure 5). The beginning of the second millennium CE saw a further, drastic drop in the intensity of human activity in this region, consistent with historical documentation on the deterioration of the Yutian Kingdom [66]. The refined chronology of the AKKQKRK site (Figure 3c) demonstrates the existence of human activity in the tailwater zone of the southern Tarim Basin during the Ming Dynasty. The two gaps in the SPD curve in the Song–Yuan and Qing Dynasty periods (Figure 5) resulted from a lack of well-preserved archaeological sites and radiocarbon dates, which require further studies to replenish. Therefore, we encourage archaeological researchers and those in other relevant spheres to promote AMS radiocarbon dating in archaeological sites and to finely report their data in future research.

## 5. Conclusions

This study acquired 25 AMS radiocarbon dates from 10 archaeological sites in the southern Tarim Basin. By applying Bayesian chronological modeling, the replenished radiocarbon dataset, constituting 91 radiocarbon dates from 18 archaeological sites in the region, refines the chronology of some of the archaeological sites and preliminarily reveals the historical trajectory of human activity. The refined chronology of the AKKQKRK site reveals the existence of human activity during the Ming Dynasty in the tailwater zone of the Taklimakan Desert. The original dating results of the Axi fortress, in front of the Kunlun Mountains, shed new light on the settlement of the Early Iron Age pastoralists. The overall human activity in the southern Tarim Basin flourished for two thousand years from the beginning of the first millennium BCE and declined in tandem with the Yutian Kingdom. This study provides the first quantitative approach to reconstructing the pattern of historical human activity in the southern Tarim Basin. However, further archaeological studies are encouraged to provide more radiocarbon dates, replenish the dataset, and further reveal the pattern of human–environment interaction in this very region. The results of this study suggest further research on human–environment interactions in the Tarim Basin, as well as in other regions along the Silk Road. Improving the understanding of this issue could help those living in arid regions to better face the challenges of global change.

**Supplementary Materials:** The following supporting information can be downloaded at https://www.mdpi.com/article/10.3390/land13040477/s1: Table S1: Archaeological radiocarbon dataset of the southern Tarim Basin.

**Author Contributions:** Conceptualization, M.Q.; methodology, M.Q.; formal analysis, X.M. and M.Q.; investigation, X.H. and M.Q.; resources, X.H.; writing—original draft preparation, X.M. and M.Q.; visualization, M.Q.; supervision, X.H. All authors have read and agreed to the published version of the manuscript.

**Funding:** This research was funded by the Second Tibetan Plateau Scientific Expedition and Research Program, grant number 2019QZKK0601; the NSFC-INSF Joint Research Project, grant number 42261144670; the National Social Science Fund of China, grant number 21VJXG010; the European

Research Council Grant, grant number ERC-2019-AdG-883700-TRAM; and the Academician and Expert Workstation of Yunnan Province, grant number 202305AF150183.

**Data Availability Statement:** The original contributions presented in the study are included in the article and Supplementary Materials, further inquiries can be directed to the corresponding author.

**Acknowledgments:** We thank Zhilin Shi, Huajie Zhang, Aili, Aisanjiang, Kayimu, and Maimaiti Awuti for their help in field work. We thank Guanghui Dong and Huihui Cao for their help in the experiment and data analysis.

**Conflicts of Interest:** The authors declare no conflicts of interest.

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
