# Peer review of "Radiocarbon-Refined Archaeological Chronology and the History of Human Activity in the Southern Tarim Basin"

_land, doi:10.3390/land13040477_

Round 1
Reviewer 1 Report
Comments and Suggestions for Authors
Xinjiang is the core area for cultural exchange between China and the West. In recent years, archaeologists and geographers have done a lot of work on the central part of Xinjiang, especially on the northern foothills of the Tianshan Mountains, and have basically reconstructed ancient human activities in this area. However, we are still not clear about the human activities in the southern margin of the Tarim Basin which is also equally important with the northern foothills of the Tianshan Mountains. This article provides 25 radiocarbon dating, which is important for understanding the mechanisms of human activities in arid regions represented by the region. This article should be published except for some minor modifications.
1. The review of research in the region has overlooked some literature (lines 48-53), such as:
Xu DK, Li C, Jin YY, 2023. Relationship between the rise and fall of Loulan ancient city and centennial-scale climate events and cycles. Frontiers of Earth Science.
Li YQ, Storozum M, Li Hm, et al., 2021. Architectural connections between western Central Asia and China: new investigations at Haermodun (cal AD 90–321), a fortified circular settlement in Xinjiang, China. Antiquty.
2. I strongly agree that the place names in the article should be in English, but their Chinese (pinyin) names should also be noted. For example: Kashgar (Kashi), Khotan (Yutian). Especially when two different writing styles are used in a picture, Khotan in Fig 1a, while Yutian in Fig.1b
3. I strongly recommend that the order of the numbers in Figure 1b be from left to right and from top to bottom. The current chaotic arrangement is extremely unfriendly to readers.
4. The author said that the carbon-14 dating data was concentrated during the Sui and Tang dynasties in line 158, but there was no in-depth explanation.
Author Response
Dear esteemed reviewer,
Thank you so much for your recommendation for a possible publication by the journal Land and constructive comments for us to finalize our manuscript. We have carefully checked all the issues raised by you and have made some modifications accordingly. Revision notes are given, point-by-point, as follows:
Point 1: Xinjiang is the core area for cultural exchange between China and the West. In recent years, archaeologists and geographers have done a lot of work on the central part of Xinjiang, especially on the northern foothills of the Tianshan Mountains, and have basically reconstructed ancient human activities in this area. However, we are still not clear about the human activities in the southern margin of the Tarim Basin which is also equally important with the northern foothills of the Tianshan Mountains. This article provides 25 radiocarbon dating, which is important for understanding the mechanisms of human activities in arid regions represented by the region. This article should be published except for some minor modifications.
Response 1: We are deeply indebted to you for highlighting the importance of refining Xinjiang’s archaeological chronology. And the recommendation for publication is highly appreciated. We have addressed all the issues raised by you and marked them out in yellow in our resubmitted manuscript.
Point 2: 1. The review of research in the region has overlooked some literature (lines 48-53), such as:
Xu DK, Li C, Jin YY, 2023. Relationship between the rise and fall of Loulan ancient city and centennial-scale climate events and cycles. Frontiers of Earth Science.
Li YQ, Storozum M, Li Hm, et al., 2021. Architectural connections between western Central Asia and China: new investigations at Haermodun (cal AD 90–321), a fortified circular settlement in Xinjiang, China. Antiquty.
Response 2: We thank you for raising this issue and would like to explain our way of addressing it part by part. A careful recheck of the literature shows that the up-to-date paper about Loulan (Xu et al., 2023 FoES) has not been overlooked in this manuscript. It was, in fact, cited as [82] in a more proper context at line 296. As suggested by you, the archaeological finding at Haermodun (Li et al., 2021 Antiquity) has now been supplemented to the review of relevant research at line 53 in the revised manuscript.
Point 3: 2. I strongly agree that the place names in the article should be in English, but their Chinese (pinyin) names should also be noted. For example: Kashgar (Kashi), Khotan (Yutian). Especially when two different writing styles are used in a picture, Khotan in Fig 1a, while Yutian in Fig.1b
Response 3: Many thanks to the suggestion and sorry for the confusion. We have made a modification accordingly about Figure 1b by labelling Khotan on the map to prevent misunderstanding. Though the region of Khotan (or Hetian) is famous for the ancient kingdom named Yutian, these two are not exactly the same place. Besides, all the cities marked out in Figure 1 are labeled with contemporary standard names. The name “Yutian” in Figure 1b represents for the Yutian County of the Khotan Region, and the name “Hetian” in Figure 1b represents for the Hetian County of the Khotan Region.
Point 4: 3. I strongly recommend that the order of the numbers in Figure 1b be from left to right and from top to bottom. The current chaotic arrangement is extremely unfriendly to readers.
Response 4: This valuable suggestion is highly appreciated. Considering your suggestion, we have rearranged the numbers of the archaeological sites in Figure 1b from right to left and from top to bottom in our revised manuscript to make them less chaotic and more reader friendly.
Point 5: The author said that the carbon-14 dating data was concentrated during the Sui and Tang dynasties in line 158, but there was no in-depth explanation.
Response 5: We are very sorry for our negligence of an in-depth explanation. We have supplemented an explanation starting from line 160 to clarify the indication of the 14C data in the resubmitted manuscript. Your great insight is very much appreciated.
Reviewer 2 Report
Comments and Suggestions for Authors
In the article "Radiocarbon-refined Archaeological Chronology and Human Activity History of the Southern Tarim Basin," the authors present 25 new radiocarbon dating results from 10 archaeological sites in the southern Tarim Basin. They combine this new chronology data with previously reported dates in a Bayesian model to explore the patterns of human-environment interaction in the region. Undoubtedly, the data provided in this research significantly contributes to the underinvestigated southern Tarim Basin. However, the contextualization of the new data by the authors is less satisfying, prompting my recommendation for publication with major revisions to address the following concerns.
The primary issue lies in the methodology. Reconstructing over 3000 years of human activity in the southern Tarim Basin with only 80 radiocarbon dates may not be entirely valid. While it is acknowledged that increasing the number of dates significantly may be impractical, the authors should avoid making potentially biased arguments. For instance, the identification of the Tang dynasty as a peak in human activity seems influenced by the authors' focus on sampling and dating many Tang dynasty sites. This choice may be due to a preference for easily accessible surface remains, neglecting other types of sites such as burials. Similarly, the identification of a small peak in Ming-Qing dynasty activity could be influenced by the authors reporting dates from this period, potentially leading to an inaccurate representation.
Furthermore, the article faces another major problem. Despite the initial intent to study human-environment interaction in the southern Tarim Basin, the authors do not seem to achieve significant new insights on this issue. This is particularly evident in the conclusion, which lacks comments on past human-environment interactions in the region. While some comments on this matter are scattered throughout the main text, the authors should synthesize their opinions to construct a coherent and convincing argument.
The Bayesian models constructed by the authors conglomerate all available dates without considering their contextual details and interrelationships. This approach results in the loss of valuable information for refining the chronology. It is suggested that the authors build a more comprehensive model that accounts for the contexts and interrelationships between dates.
Additionally, there are several specific problems that should be addressed:
- The connection made between the drastic drop in human activities and the fall of the Yutian Kingdom lacks strong evidence. The authors are advised to avoid misleading statements without substantial support.
- The term "representative sites" needs clarification. The authors should explain the criteria used to judge representativeness and justify why these specific 25 sites were sampled.
- In Line 59, the term "independent" requires clarification, and the authors should consider elaborating on the interdependencies of oases in the north.
- Figure 1 should use a different color scheme to represent different periods, using shades of red to denote quantitative differences.
- Proper references are needed to support the argument in Line 262-265, as the cited article cannot prove the "synchronicity."
- In Line 252, it should be Figure 4d rather than Figure 4B. Consistency in figure labels (Figure 1,2,3,4a/b/c/d) throughout the article is recommended.
- References for each date listed in the supplementary spreadsheet should be added to facilitate data verification.
Finally, the language in the manuscript needs significant improvement. Authors are encouraged to carefully read the manuscript multiple times to rectify language problems. Specific issues, such as the repetition of the sentence "we carried out a field investigation in 2018–2019" in Line 123, should be addressed.
Minor problems identified line by line:
- Line 72: The phrase "sedentary human activity" is awkward. Consider replacing it with something like "sedentary population."
- Line 263: The term "steppe" refers to the Eurasian Steppe. Refrain from calling the grassland in the foothills of the Kunlun Mountains the steppe.
The English language of this article can be significantly improved with the help of a native speaker.
Author Response
Response to Reviewer 2 Comments
Dear esteemed reviewer,
We are deeply indebted to you for reviewing our manuscript so carefully and giving us detailed so many professional and valuable comments on it. All the suggestions you have given us are not only constructive in the finalization of this manuscript, but also helpful for our future publications. We have tried our best to improve the manuscript and have made some modifications according to your kind suggestions in our revised manuscript. Hopefully, this could be acceptable to you and the journal, but we are happy to conduct further revisions if necessary.
Point 1: In the article "Radiocarbon-refined Archaeological Chronology and Human Activity History of the Southern Tarim Basin," the authors present 25 new radiocarbon dating results from 10 archaeological sites in the southern Tarim Basin. They combine this new chronology data with previously reported dates in a Bayesian model to explore the patterns of human-environment interaction in the region. Undoubtedly, the data provided in this research significantly contributes to the underinvestigated southern Tarim Basin. However, the contextualization of the new data by the authors is less satisfying, prompting my recommendation for publication with major revisions to address the following concerns.
Response 1: We are so grateful to you for the recommendation for publication and so sorry for the poor contextualization of the new data found in this study. We have carefully addressed this issue in our resubmitted version of the paper. Please see how we have revised the paper following your honest and kind suggestions. Thank you again.
Point 2: The primary issue lies in the methodology. Reconstructing over 3000 years of human activity in the southern Tarim Basin with only 80 radiocarbon dates may not be entirely valid. While it is acknowledged that increasing the number of dates significantly may be impractical, the authors should avoid making potentially biased arguments. For instance, the identification of the Tang dynasty as a peak in human activity seems influenced by the authors' focus on sampling and dating many Tang dynasty sites. This choice may be due to a preference for easily accessible surface remains, neglecting other types of sites such as burials. Similarly, the identification of a small peak in Ming-Qing dynasty activity could be influenced by the authors reporting dates from this period, potentially leading to an inaccurate representation.
Response 2: Many thanks to the critical comment. We highly agree with you that the more sufficient the radiocarbon data are, the more accurate the reconstruction of human activity will be. However, restricted by the scarcity of the radiocarbon data published till now, this approach can only expect to provide a preliminary understanding on this significant issue based on the data available. We have checked and revised the manuscript to make sure that all the statements are strictly supported by the results and credible evidences. Biases in site and sample selection during the field investigation and dating process is consciously avoided, which means that the surficial archaeological remains were evenly treated. The concentration of radiocarbon dates during Tang Dynasty and Ming-Qing period in southern Tarim Basin is consistent with previous results in Xinjiang (e.g., Cao et al., 2022; Ding et al., 2023). Hence, we believe that the trend revealed by the reconstruction in this study is reliable, though future supplementary of new radiocarbon dates might help to make the pattern more accurate. We have added a statement in lines 324–328 to restrict our conclusions following your suggestion. Please see similar reports in publications as follows:
Cao, H.H.; Wang, Y.Q.; Qiu, M.H.; Shi, Z.L.; Dong, G.H. On the exploration of social development during a historical period in the Eastern Tianshan Mountains via archaeological and geopolitical perspectives. Land 2022, 11, 1416.
Ding, G.Q.; Chen, J.H.; Lei, Y.B.; Lv, F.Y.; Ma, R.; Chen, S.Q.; Ma, S.; Sun, Y.H.; Li, Y.C.; Wang, H.P.; Shi, Z.L.; Seppä, H.; Chen, F.H. Precipitation variations in arid central Asia over past 2500 years: Possible effects of climate change on development of Silk Road civilization. Global Planet. Change 2023, 226, 104142.
Point 3: Furthermore, the article faces another major problem. Despite the initial intent to study human-environment interaction in the southern Tarim Basin, the authors do not seem to achieve significant new insights on this issue. This is particularly evident in the conclusion, which lacks comments on past human-environment interactions in the region. While some comments on this matter are scattered throughout the main text, the authors should synthesize their opinions to construct a coherent and convincing argument.
Response 3: Thank you so much for this constructive comment concerning this critical issue. As we have stated in the abstract, introduction and other parts of the manuscript, this paper aimed to preliminarily reconstruct a local human activity history of the southern Tarim Basin, which could be fundamental for future work focusing on human-environment interaction. Previous researches focusing on this issue have been carefully reviewed in this study. However, in the absence of new results from palaeoenvironment reconstruction, we can only come to a very cautious conclusion based on the data collected. Following your valuable suggestion, we have added our comments on past human-enviornment interactions in the region and synthesized our opinions to make our argument more coherent and convincing in the conclusion part. Consistent with your expectations, we are looking forward to improving the understanding of this important issue in future studies.
Point 4: The Bayesian models constructed by the authors conglomerate all available dates without considering their contextual details and interrelationships. This approach results in the loss of valuable information for refining the chronology. It is suggested that the authors build a more comprehensive model that accounts for the contexts and interrelationships between dates.
Response 4: We appreciate your valuable suggestion and totally agree with you that taking contexts and interrelationships into consideration is important to refine the chronology. Nevertheless, the new radiocarbon dates yield in this study are from field investigation, instead of systematic archaeological excavation, and the quantity of the data is restricted. Some of the previous data are also facing two major challenges: 1) poor data quality, and 2) unclearly identified stratigraphy and/or context. We believe that more detailed archaeological researches and chronological reports are significant and much needed to refine the chronology of the region. Therefore, my co-authors and I will endeavor to contribute our due part concerning this.
Point 5: Additionally, there are several specific problems that should be addressed:
- The connection made between the drastic drop in human activities and the fall of the Yutian Kingdom lacks strong evidence. The authors are advised to avoid misleading statements without substantial support.
- The term "representative sites" needs clarification. The authors should explain the criteria used to judge representativeness and justify why these specific 25 sites were sampled.
- In Line 59, the term "independent" requires clarification, and the authors should consider elaborating on the interdependencies of oases in the north.
- Figure 1 should use a different color scheme to represent different periods, using shades of red to denote quantitative differences.
- Proper references are needed to support the argument in Line 262-265, as the cited article cannot prove the "synchronicity."
- In Line 252, it should be Figure 4d rather than Figure 4B. Consistency in figure labels (Figure 1,2,3,4a/b/c/d) throughout the article is recommended.
- References for each date listed in the supplementary spreadsheet should be added to facilitate data verification.
Response 5: We are grateful to you for pointing out these specific issues. We have carefully addressed all of them in the revised manuscript. Revision notes are given, point-by-point, as follows:
- Thank you for this critical comment and sorry for the possible misunderstanding. The result of radiocarbon dating in this paper has shown a drastic decline since the beginning of the second millennium CE, which is consistent with the historical record of the perish of the ancient Yutian Kingdom in 1006 CE. In the manuscript, we only dare to make statements based on the data collected to describe this solid phenomenon instead of attempting to extend to any risky conclusions without sound evidence.
- Thanks for pointing this out. We have replaced the term “representative site” with “undated site” in the resubmitted manuscript to improve accuracy.
3.We apologize for the confusion generated by our lack of consideration when textualizing our opinions and thanks for your suggestion. We have added the term “mutually” to restrict the meaning of the term “independent” in the revised manuscript. As for the oases in the north, the first half of the sentence at line 58 has declared that these oases are large and strung by the Tarim River.
- Your kind suggestion is highly appreciated. We have made some modification on Figure 1 to improve clarity following your suggestion. What we need to explain to you is that different periods have already been represented by different colours in this figure, and each spot represents a single archaeological site, which means that quantitative difference is not involved in the expression of the figure.
5.We are very grateful to you for pointing this out and sorry for the misstatement. Considering your suggestion, we have modified the sentence at lines 269–272 in the resubmitted version of the manuscript to precisify the statement.
- Many thanks for pointing out this mistake. We have corrected the citation in the revised manuscript according to your suggestion. What we need to clarify to you is that for the consistency of figure labels, we have formatted this manuscript following the template provided by the journal, which indicates lower cases in the figures and upper cases in the citation. However, we are always willing to make further adjustments if the editor asks.
- We agree with this suggestion and have modified the supplementary table to realize consistency with the manuscript as you have suggested, and we confirm now that all references involved in the supplementary material are listed in the Reference section in the resubmitted manuscript.
Point 6: Finally, the language in the manuscript needs significant improvement. Authors are encouraged to carefully read the manuscript multiple times to rectify language problems. Specific issues, such as the repetition of the sentence "we carried out a field investigation in 2018–2019" in Line 123, should be addressed. Minor problems identified line by line:
- Line 72: The phrase "sedentary human activity" is awkward. Consider replacing it with something like "sedentary population."
- Line 263: The term "steppe" refers to the Eurasian Steppe. Refrain from calling the grassland in the foothills of the Kunlun Mountains the steppe.
Response 6: We apologize for the poor language of our manuscript. Many thanks for helping us improve the quality of this manuscript. We have carefully addressed these language issues following your valuable suggestions and dedicated to improve the language by reviewing the whole manuscript for several times to improve readability and made some modifications with the help of one native English speaker, James Greeland from England. All modifications are labeled in yellow in the resubmitted manuscript. Your insightful evaluation and constructive feedback are very much appreciated, we just can’t thank you enough.

Reviewer 3 Report
Comments and Suggestions for Authors
The origin of the cartographic bases in figure 1 remains to be indicated. The authorship of the remaining figures and tables must also be clearly specified. In figure 2, photographs should be indicated instead of photograph.
The conclusions refer to 10 archaeological sites, but in the work up to 18 are handled. The conclusions are very brief and must be developed further, drawing conclusions not only with respect to pure chronology, but with respect to the different typologies of the archaeological sites studied and to its location in the different scenarios studied in the Southern Tarim Basin. It is recommended to appreciate the density of settlements in the different chronological sections observed, taking into account, as far as possible, the evolution of the environment.
Author Response
Response to Reviewer 3 Comments
Dear esteemed reviewer,
On behalf of all the contributing authors, I would like to express our sincere appreciations for your kind feedback and constructive comments on our manuscript. These insightful comments are all valuable and helpful for improving and finalizing our article. We have carefully checked all the issues raised by you and have made some modifications following your suggestions. Revision notes are given, point-by-point, as follows:
Point 1: The origin of the cartographic bases in figure 1 remains to be indicated. The authorship of the remaining figures and tables must also be clearly specified. In figure 2, photographs should be indicated instead of photograph.
Response 1: You have our heart-felt gratitude for evaluating this work and providing constructive comments. We have now clearly indicated the cartographic bases of Figure 1 as well as the authorship of the remaining figures and tables according to your kind suggestions. The word “photographs” has been modified in Figure 2 as you have suggested as well in the resubmitted version of our manuscript. One point to clarify, all of the photographs were shot by the corresponding author himself.
Point 2: The conclusions refer to 10 archaeological sites, but in the work up to 18 are handled. The conclusions are very brief and must be developed further, drawing conclusions not only with respect to pure chronology, but with respect to the different typologies of the archaeological sites studied and to its location in the different scenarios studied in the Southern Tarim Basin. It is recommended to appreciate the density of settlements in the different chronological sections observed, taking into account, as far as possible, the evolution of the environment.
Response 2: Thank you so much for your professional suggestions concerning the Conclusion part. Considering your comments, we have supplemented a few sentences to the relevant paragraph, please see lines 315–316 and 324–328. This paper aims to report some new AMS radiocarbon dates and raise the issue of insufficient radiocarbon dating work of the southern Tarim Basin. We strongly agree with you that the typologies of the archaeological sites and their densities should be considered to better unveil the trajectory of human activities in this region. However, a more comprehensive investigation to finely cover the archaeological remains and more precise chronologies of these remains are yet to be conducted. We hope that future researches can address these issues and fill these blanks soon.

Round 2
Reviewer 2 Report
Comments and Suggestions for Authors
The revised version has addressed all the issues I pointed out. I recommend the publication of this manuscript as it is now.
Author Response
Response to Reviewer 2 Comments
Point 1: The revised version has addressed all the issues I pointed out. I recommend the publication of this manuscript as it is now.
Response 1: We are very glad to receive the recommendation for publication. Thanks again for the efforts the reviewer has contributed to this manuscript.
